# Subtypes of Patients with Mild to Moderate Airflow Limitation as Predictors of Chronic Obstructive Pulmonary Disease Exacerbation

**DOI:** 10.3390/jcm12206643

**Published:** 2023-10-20

**Authors:** Nam Eun Kim, Eun-Hwa Kang, Ji Ye Jung, Chang Youl Lee, Won Yeon Lee, Seong Yong Lim, Dong Il Park, Kwang Ha Yoo, Ki-Suck Jung, Jin Hwa Lee

**Affiliations:** 1Division of Pulmonary and Critical Care Medicine, Department of Internal Medicine, Ewha Womans University Seoul Hospital, Ewha Womans University College of Medicine, Seoul 07804, Republic of Korea; gnikcor88@gmail.com; 2Informatization Department, Ewha Womans University Medical Center, Seoul 07985, Republic of Korea; eeunhwak@gmail.com; 3Division of Pulmonary and Critical Care Medicine, Department of Internal Medicine, Severance Hospital, Yonsei University College of Medicine, Seoul 06273, Republic of Korea; stopyes@yuhs.ac; 4Division of Pulmonary, Allergy and Critical Care Medicine, Hallym University Chuncheon Sacred Heart Hospital, Chuncheon 24253, Republic of Korea; doclcy@gmail.com; 5Division of Pulmonary, Allergy and Critical Care Medicine, Department of Internal Medicine, Yonsei University Wonju College of Medicine, Wonju 26426, Republic of Korea; wonylee@yonsei.kr; 6Division of Pulmonary and Critical Care Medicine, Department of Medicine, Kangbuk Samsung Hospital, Sungkyunkwan University School of Medicine, Seoul 03181, Republic of Korea; mdlimsy@skku.edu; 7Division of Pulmonary and Critical Care Medicine, Department of Internal Medicine, Chungnam National University Hospital, Daejeon 35015, Republic of Korea; rahm3s@gmail.com; 8Division of Pulmonary and Allergy, Department of Internal Medicine, Konkuk University Hospital, School of Medicine, Konkuk University, Seoul 05030, Republic of Korea; 20010025@kuh.ac.kr; 9Division of Pulmonary, Allergy and Critical Care Medicine, Department of Internal Medicine, Hallym University Sacred Heart Hospital, Anyang 14068, Republic of Korea; pulmoks@hallym.ac.kr

**Keywords:** subtypes of COPD, acute exacerbation, k-means clustering, multicenter observation cohort study

## Abstract

COPD is a heterogeneous disease, and its acute exacerbation is a major prognostic factor. We used cluster analysis to predict COPD exacerbation due to subtypes of mild–moderate airflow limitation. In all, 924 patients from the Korea COPD Subgroup Study cohort, with a forced expiratory volume (FEV1) ≥ 50% and documented age, body mass index (BMI), smoking status, smoking pack-years, COPD assessment test (CAT) score, predicted post-bronchodilator FEV1, were enrolled. Four groups, putative chronic bronchitis (n = 224), emphysema (n = 235), young smokers (n = 248), and near normal (n = 217), were identified. The chronic bronchitis group had the highest BMI, and the one with emphysema had the oldest age, lowest BMI, and highest smoking pack-years. The young smokers group had the youngest age and the highest proportion of current smokers. The near-normal group had the highest proportion of never-smokers and near-normal lung function. When compared with the near-normal group, the emphysema group had a higher risk of acute exacerbation (OR: 1.93, 95% CI: 1.29–2.88). However, multiple logistic regression showed that chronic bronchitis (OR: 2.887, 95% CI: 1.065–8.192), predicted functional residual capacity (OR: 1.023, 95% CI: 1.007–1.040), fibrinogen (OR: 1.004, 95% CI: 1.001–1.008), and gastroesophageal reflux disease were independent predictors of exacerbation (OR: 2.646, 95% CI: 1.142–6.181). The exacerbation-susceptible subtypes require more aggressive prevention strategies.

## 1. Introduction

Chronic obstructive pulmonary disease (COPD) is the third major cause of death worldwide, and its prevalence is expected to rise over the next 30 years to up to 4.5 million cases by 2030, while increasing the risk of death and healthcare costs [1,2].

COPD is traditionally defined as the post-bronchodilator limitation of expiratory airflow, which is indicated by a forced expiratory volume in 1 s (FEV1)/a forced vital capacity (FVC) < 0.7 [3]. However, COPD is a heterogeneous group of many diseases with different causes, mechanisms, and structural abnormalities that arise as lung function declines with age [4,5]. Recent studies have described a variable lung function trajectory of accelerated FEV1 decline caused by early-life disadvantages and exposure to risk factors, especially smoking, and also highlighted the importance of low FEV1 in early adulthood as an indicator of early COPD [6,7,8]. However, it is hard to detect early COPD in the current clinical settings, and it can remain underdiagnosed based on the current COPD definition based on the FEV1/FVC ratio [9].

Based on its natural history, early COPD is more likely to progress and be diagnosed as late, mild–moderate, or severe during middle age or later rather than to remain stable without progressing. Because early-COPD patients rarely seek medical attention for preclinical symptoms, it is difficult to prevent the progression of early COPD, for example, by quitting smoking. Colak et al. reported that early-COPD patients had significantly higher hospitalization and mortality rates over a 14-year follow-up, highlighting the importance of managing early COPD [10].

The category of patients with mild–moderate airflow limitation may include patients who have progressed from early to late COPD, those with mild–moderate obstruction, and patients diagnosed with COPD because of accelerated lung function decline. Of the patients in this group, some progress and develop severe airflow limitation, which leads to a poor prognosis. Therefore, a better understanding of the heterogeneity of patients with mild–moderate airflow limitation may have significant clinical value because it is easier to diagnose than early COPD, and its diagnosis may provide the opportunity to alter the course of the disease before it develops into severe COPD.

The k-means clustering algorithm learns groups or patterns of data using a function called k-means in the R-based library environment. The advantage of k-means clustering is that it provides researchers with a reasonable-similarity group by obtaining a qualitative and quantitative understanding of a large amount of N-dimensional data [11]. Previous studies have also used clustering for data-driven patient grouping in medical research [12,13].

In this study, we use k-means clustering to subgroup mild to moderate COPD patients and distinguish the phenotypes of four groups, as “chronic bronchitis”, “emphysema”, “young smokers”, and “near normal” and predict its prognosis. The data were drawn from the Korea COPD Subgroup Study (KOCOSS) cohort.

## 2. Materials and Methods

### 2.1. Data Collection

The KOCOSS cohort is a multicenter observation cohort study that enrolled COPD patients from 45 Korean tertiary and university hospitals from December 2011 to October 2014. The inclusion criteria were COPD diagnosis by a pulmonologist, being ≥40 years old, COPD symptoms (e.g., coughing, sputum, and dyspnea), and a post-bronchodilator FEV1/FVC value of <0.7 [3,14]. The medical history taken at the first hospital visit included the frequency and severity of exacerbations in the previous 12 months, smoking status, treatment (including previously prescribed COPD medications), and comorbidities. All of the data were reported using case-report forms (CRFs) completed by physicians or trained nurses, and the patients were to be evaluated at regular 6-month intervals after the initial examination.

The main exclusion criteria were asthma, bronchiectasis and tuberculosis lung damage included in obstructive lung disease, incapability to complete the pulmonary function test, myocardial infarction or a cerebrovascular event in the previous three months, pregnancy, rheumatoid disease, malignancy, inflammatory bowel disease, and steroid use for conditions other than COPD exacerbation within eight weeks before enrollment [14].

The pulmonary function, disease severity, and exercise capacity were examined using spirometry and the six-minute walk test (6MWT), following standard procedures [15,16].

The smoking status was defined as never (someone who had smoked less than 100 cigarettes in their lifetime and did not currently smoke), former (someone who had smoked at least 100 cigarettes in their lifetime but had not smoked in the previous month), or current smokers (someone who had smoked at least 100 cigarettes in their lifetime and had smoked within the previous month) [17].

COPD exacerbations were defined as acute respiratory symptoms requiring either antibiotics or systemic steroids and severe respiratory events requiring hospitalization [18].

### 2.2. Study Population

Of the 2181 subjects in the KOCOSS cohort, 93 with missing baseline pulmonary function test data, 588 with insufficient follow-up data, and 532 with measured post-bronchodilator FEV1 values of <50% were excluded from the analyses.

The variables age, body mass index (BMI), smoking status, smoking amount, predicted post-bronchodilator FEV1 (%), and COPD assessment test (CAT) score were selected for cluster analysis (Figure 1). Forty-four subjects, who were missing at least one of the six key input variables, were excluded. Our cluster analysis of COPD subtypes included 924 patients, and it identified four groups: Cluster 1 (n = 224), Cluster 2 (n = 235), Cluster 3 (n = 248), and Cluster 4 (n = 217), which were named the chronic bronchitis group, the emphysema group, the young smokers group, and the near-normal group, respectively (Figure 2).

### 2.3. Statistical Analysis

A k-means clustering analysis was conducted on a subset of KOCOSS subjects with complete data on the six variables. We used normalized mutual information to determine the optimal number of clusters. The “NbClust” machine learning library was used to determine the appropriate number of clusters for clustering [19]. After clustering, the clinical characteristics were compared across the groups. The continuous variables are presented as mean ± standard deviation. The categorical variables are presented as proportions (%). One-way ANOVA or the Kruskal–Wallis test was used to analyze continuous variables. The Pearson chi-square test or the Fisher’s exact test was used to analyze categorical variables. Acute exacerbation risk factors, including COPD subtypes, were analyzed using multiple logistic regression analysis and reported as odds ratios (ORs) with 95% confidence intervals (CIs). Based on univariate analysis results, variables with *p* < 0.1 were subjected to multivariate analysis. To handle the multicollinearity of correlated independent variables, we excluded variables with a variable inflation factor of >10. Furthermore, *p* < 0.05 (two-tailed) indicated statistically significant differences. All statistical analyses were performed on SPSS version 24.0 (SPSS Inc., Chicago, IL, USA).

## 3. Results

### 3.1. Baseline Characteristics

Clustering analysis was used to group the 924 subjects included in this study into the putative chronic bronchitis (n = 224), emphysema (n = 235), young smokers (n = 248), and near-normal (n = 217) groups.

Table 1 shows the baseline characteristics of the subjects based on the cluster groups. The proportion of males was high in all groups. Of the four groups, the chronic bronchitis group had the highest mean BMI, whereas the emphysema group had the highest mean age, lowest BMI, and the highest smoking pack-years (all *p* < 0.001). The young smokers group had the lowest mean age and the highest number of current smokers. The near-normal group had the highest proportion of never-smokers and the lowest number of current smokers (all *p* < 0.001).

Pulmonary function analysis revealed that the near-normal group had the best lung function, and their mean FEV1 was above 80% of the normal predicted value. The chronic bronchitis and emphysema groups had the lowest predicted post-bronchodilator FEV1 (%), and the chronic bronchitis group had the lowest predicted post-bronchodilator FVC (%) (*p* < 0.001), whereas the emphysema group showed the lowest diffusing capacity (DLCO) (*p* < 0.001).

The median CAT score and the St. Georges Respiratory Questionnaire (SGRQ) score were highest in the emphysema group (all *p* < 0.001). Based on the 6MWT, the emphysema group had the highest dyspnea score (*p* < 0.001) and the shortest distance walked (*p* < 0.001).

Although not all subjects had chest computed tomography (CT) results, emphysema was most frequent in the emphysema group (*p* = 0.017), whereas bronchiectasis was most common in the chronic bronchitis group (*p* = 0.002, Appendix A).

### 3.2. Comorbidities and Medical History

Among the four groups, the proportion of diabetes mellitus, hypertension, and asthma was the highest in the chronic bronchitis group (all *p* < 0.001, Table 2).

The most commonly prescribed medications, regardless of COPD clusters, were long-acting muscarinic antagonists, followed by a combination of inhaled corticosteroids (ICSs) and long-acting beta-2 agonists (LABAs). The use of methylxanthine was the highest in emphysema group (*p* < 0.001, Table 3).

### 3.3. Occurrence of Acute Exacerbation and Mortality

The number of exacerbations during one year prior to the first visit was significantly higher in the emphysema group (*p* < 0.001). In case of severe exacerbation, there was no significant difference between the groups. Mortality was the highest in the emphysema group (*p* < 0.05, Table 4).

Among the four groups, the emphysema group was associated with a greater risk of acute exacerbation when compared to the near-normal group (OR, 1.93, 95% CI, 1.29 to 2.88, *p* < 0.001) (Figure 3).

A multiple logistic regression analysis was performed including non-clustered variables with a *p*-value < 0.1 in univariate analysis (Appendix A). As a result, for the chronic bronchitis group (OR, 2.887; 95% CI, 1.065–8.192, *p* = 0.040), functional residual capacity (FRC) % predicted (OR, 1.023; 95% CI, 1.007–1.040, *p* = 0.007), fibrinogen (OR, 1.004; 95% CI, 1.001–1.008, *p* = 0.015), and gastroesophageal reflux disease (GERD) were independent predictors of exacerbation in patients with mild to moderate airflow limitation (OR, 2.646; 95% CI,1.142–6.181, *p* = 0.023) (Table 5).

## 4. Discussion

This study subtyped patients with a mild to moderate airflow limitation through cluster analysis using variables closely related to the prognosis of COPD. Our emphysema group showed characteristics such as the oldest age, the lowest BMI, the highest pack-years of smoking over 50 pack-years, the second-lowest FEV1 % predicted, and the highest CAT score. In this group, emphysema was observed in almost half of the cases for which a chest CT was available. Predictably, the emphysema group had more moderate exacerbations and higher mortality than the other groups. In particular, when compared with the near-normal group, which had the highest ratio of never-smokers and the best lung function, only the emphysema group had a significantly higher risk of exacerbation. However, it was not an independent predictor of exacerbation in the results of the multiple logistic regression analysis.

Our chronic bronchitis group appeared to be independently susceptible to exacerbation in the multiple logistic regression analysis model. The chronic bronchitis group had the lowest FEV1 and the highest BMI and had many findings of bronchiectasis on chest CT, suggesting similar findings to chronic bronchitis, which is commonly described as one of the clinical phenotypes of COPD. In addition, the chronic bronchitis group in our study had a high prevalence of diabetes mellitus, hypertension, and asthma, which are well known to be related to a high BMI. It is likely that these comorbidities accelerated inflammation in the chronic bronchitis group, making them more susceptible to acute exacerbations. In previous studies by Kim et al., the chronic bronchitis group was younger, had lower FEV1, had more males, had more current smokers, and had a higher exacerbation rate than those without chronic bronchitis [20,21]. From our chronic bronchitis group, we too infer that chronic bronchitis features are predictors of COPD exacerbations even in patients with mild to moderate airflow limitation, even though the emphysema group, unlike the chronic bronchitis group, was the oldest, had the lowest BMI, and had the lowest quality of life. Furthermore, FRC [22], fibrinogen [23,24], and GERD [25,26,27], previously known as predictors of the exacerbation of COPD, were also identified as independent predictors in our study, indicating the validity of our cluster analysis results.

The clinical meaning of acute exacerbation is important because it is not just an acute, worsening respiratory event but an event that leads to more frequent non-respiratory adverse events [28] and a relapse of acute exacerbation with hospitalization, healthcare costs [29], and finally COPD-related mortality [30]. Previous studies have reported the clinical importance of acute exacerbation in mild COPD [25]. Dransfield et al. reported that exacerbation is associated with greater lung function decline in subjects with COPD, especially those with mild COPD (GOLD stage 1 or 2), than in those with GOLD stage 3 or 4. In the GOLD 1 group, the additional FEV1 decline was 23 mL/yr; otherwise, it was 10 mL/yr in GOLD 2, 8 mL/yr in GOLD 3, and no change in GOLD 4, respectively. This implies that mild COPD can potentially progress to more advanced lung disease when confronted with acute exacerbation [31].

Through this study, we hypothesis that, among COPD clusters, a specific cluster could be susceptible to acute exacerbation leading to pulmonary function decline, reducing the quality of life and resulting in increasing mortality. These results are useful for the early detection of high-risk subtypes susceptible to exacerbation and for drug use to reduce the risk of acute exacerbation and mortality. For example, if medical resources are limited and only a subset of COPD patients with mild to moderate airflow restriction should be selected with priority, patients with chronic bronchitis features with a low FEV1 or emphysematous feature with a low FEV1 and a poor quality of life should be prioritized. For these patients, pharmacologic interventions such as mucous removal or inhalers and education on smoking cessation should be conducted more actively.

The strength of this study is that we identified which clinical phenotype easily causes exacerbation in patients with mild to moderate airflow limitation. The previous literature has suggested some types of COPD that were susceptible to exacerbation independent of disease severity [25], but our study has the advantage of dividing the groups more systematically through a clustering technique based on variables that can be easily confirmed clinically or by interview. The second advantage of our study is that it used a large representative COPD cohort. The KOCOSS cohort is composed of a relatively large number of patients with mild to moderate airflow limitation (mean post-bronchodilator FEV1 55.8% predicted) [14] compared to other larger cohorts such as the genetic epidemiology of COPD cohort (COPDGene, mean FEV1 48.3% predicted) [32] or the ECLIPSE cohort (mean FEV1 48.9% predicted) [5]. In addition, numerous factors affecting acute exacerbation were comprehensively analyzed.

Our study has some limitations. First, because this study was conducted in Korea, and the sample size of 924 participants is relatively small, generalization to other ethnic groups and countries could be difficult. Also, this study focused on patients with mild to moderate airflow limitation; therefore, generalization could be difficult across the GOLD stage. Second, we provide evidence based on cross-sectional analysis of the data collected at the time of enrollment in KOCOSS, so longitudinal data are limited. Third, methylxantine is not indicated for COPD; however, it is commonly used in Korea, so data related to its use were also included in the analysis, hampering generalizability. Further, well-designed studies could aim to include larger and more diverse samples to improve the external validity of the findings.

## 5. Conclusions

COPD heterogeneity exists in patients with mild to moderate airflow obstruction, and the “chronic bronchitis” phenotype showed a high risk of exacerbation. For subtypes at a high risk of the exacerbation of COPD, active intervention efforts are needed to prevent exacerbation.

## Figures and Tables

**Figure 1 jcm-12-06643-f001:**
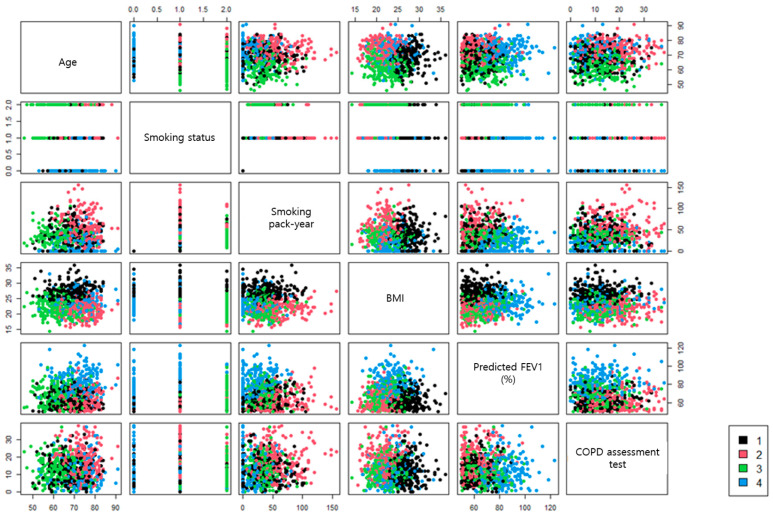
Scatterplot of cluster analysis.

**Figure 2 jcm-12-06643-f002:**
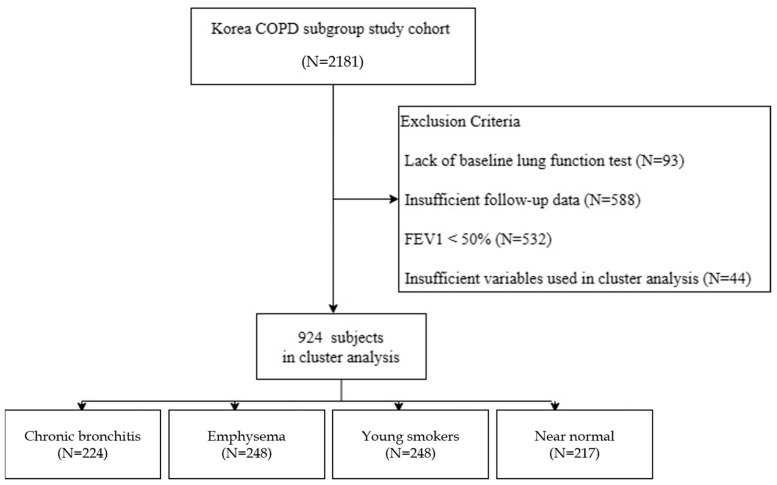
Overview of study design.

**Figure 3 jcm-12-06643-f003:**
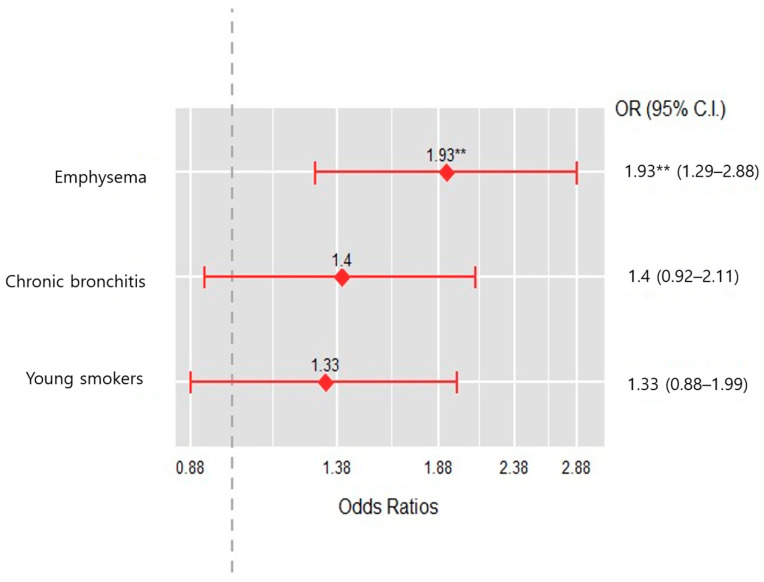
Odds ratio of moderate acute exacerbation among cluster groups. ** are presented as *p* < 0.001.

**Table 1 jcm-12-06643-t001:** Comparison of baseline characteristics according to cluster analysis groups.

	Chronic Bronchitis (N = 224)	Emphysema (N = 235)	Young Smokers (N = 248)	Near Normal (N = 217)	*p*-Value
Age, yr,	68.6 ± 6.7	74.2 ± 5.4	63.3 ± 6.6	71.9 ± 6.6	<0.001
Male	210 (93.8%)	230 (97.9%)	236 (95.2%)	177 (81.6%)	<0.001
Height, cm	165.1 ± 7.1	164.3 ± 6.2	165.5 ± 6.3	163.1 ± 7.9	0.004
Weight, kg	73.9 ± 8.8	58.6 ± 7.8	61.5 ± 8.5	62.9 ± 8.9	<0.001
BMI, kg/m^2^	27.1 ± 2.4	21.7 ± 2.3	22.4 ± 2.6	23.6 ± 2.7	<0.001
Smoking status					<0.001
Never smoker	13 (5.8%)	1 (0.4%)	0 (0.0%)	62 (28.6%)	
Former smoker	196 (87.5%)	199 (84.7%)	66 (26.6%)	152 (70.0%)	
Current smoker	15 (6.7%)	35 (14.9%)	182 (73.4%)	3 (1.4%)	
Smoking age, yr	22.6 ± 9.0	23.3 ± 12.1	22.0 ± 8.4	25.7 ± 11.9	<0.001
Smoking, pack-yrs	42.5 ± 21.8	55.3 ± 27.7	37.1 ± 16.0	29.8 ± 16.4	<0.001
Pulmonary function test					
FEV1, % predicted	62.3 ± 8.5	62.5 ± 9.3	66.9 ± 10.3	80.8 ± 12.9	<0.001
FVC, % predicted	79.4 ± 13.9	82.7 ± 13.0	88.9 ± 12.2	92.4 ± 13.6	<0.001
DLCO, % predicted	72.7 ± 19.7	58.7 ± 19.1	70.0 ± 19.6	73.8 ± 19.6	<0.001
FRC, % predicted	106.6 ± 26.9	115.1 ± 28.2	117.1 ± 24.1	102.4 ± 24.8	<0.001
RV, % predicted	95.8 ± 33.2	104.4 ± 39.1	98.8 ± 33.2	83.3 ± 33.5	<0.001
TLC, L	1.0 ± 0.2	1.0 ± 0.2	0.9 ± 0.2	0.8 ± 0.2	<0.001
Symptom scores					
CAT score	12.1 ± 6.7	17.5 ± 8.1	12.2 ± 6.5	11.4 ± 7.4	<0.001
SGRQ score	27.8 ±15.1	36.6 ± 18.9	24.8 ± 12.8	11.4 ± 7.4	<0.001
6 min walk distance, m	396.8 ± 115.9	351.7 ± 119.3	427.4 ± 99.4	388.2 ± 113.9	<0.001
Dyspnea score after 6MWT	1.7 ± 1.6	1.9 ± 1.8	1.3 ± 1.3	1.2 ± 1.3	<0.001
Laboratory findings					
WBCs, ×10^3^/mm^3^	7.4 ± 2.5	7.4 ± 2.2	7.6 ± 2.2	6.9 ± 2.1	0.002
Hb, g/dL	14..3 ± 1.7	13.8 ± 1.6	14.5 ± 1.4	14.1 ± 1.4	<0.001
ESR, mm/h	15.9 ± 17.1	20.1 ± 19.3	15.7 ± 16.4	14.6 ± 16.5	<0.001
Neutrophil, %	56.5 ± 11.7	60.7 ± 11.6	56.1 ± 10.9	57.2 ± 10.3	<0.001
Lymphocyte, %	30.3 ± 9.5	26.6 ± 9.5	31.6 ± 9.1	30.1 ± 8.4	<0.001
Monocyte, %	7.8 ± 2.4	7.6 ± 2.3	7.7 ± 4.5	8.1 ± 2.9	<0.001
Albumin, g/dL	4.4 ± 0.4	4.3 ± 0.4	4.4 ± 0.4	4.4 ± 0.4	<0.001
NT Pro-BNP, pg/mL	216.4 ± 551.5	325.5 ± 847.9	91.8 ± 167.5	118.5 ± 256.7	<0.001
D-dimer, ug/mL	0.7 ± 0.9	0.6 ± 0.5	0.5 ± 1.3	0.6 ± 0.7	0.001
Fibrinogen, mg/dL	332.9 ± 108.8	341.6 ± 101.5	324.4 ± 98.2	305.6 ± 73.8	<0.001

Data are presented as number (%) or mean ± SD. BMI, body mass index; FEV1, forced expiratory volume in 1 s; FVC, forced vital capacity; DLCO, carbon monoxide diffusing capacity; FRC, functional residual capacity; RV, residual volume; TLC, total lung capacity; CAT, COPD assessment test; SGRQ, St. George’s Respiratory Questionnaire; 6MWT, 6 min walk test; WBCs, white blood cells; Hb, hemoglobin; ESR, erythrocyte sedimentation rate; NT Pro-BNP, N-terminal pro-brain natriuretic peptide.

**Table 2 jcm-12-06643-t002:** Comparison of comorbidities according to cluster analysis groups.

	Chronic Bronchitis (N = 224)	Emphysema (N = 235)	Young Smokers (N = 248)	Near Normal (N = 217)	*p*-Value
Myocardial infarction	12 (5.4%)	16 (6.8%)	5 (2.0%)	11 (5.1%)	0.089
Heart failure	9 (4.0%)	8 (3.4%)	5 (2.0%)	6 (2.8%)	0.621
Peripheral vascular disease	5 (2.2%)	9 (3.8%)	4 (1.6%)	1 (0.5%)	0.086
Diabetes mellitus	69 (30.8%)	41 (17.5%)	41 (16.7%)	29 (13.4%)	0.000
Hypertension	124 (55.9%)	94 (40.2%)	93 (37.7%)	91 (42.1%)	0.000
Osteoporosis	10 (4.5%)	8 (3.4%)	9 (3.7%)	11 (5.1%)	0.797
GERD	38 (17.0%)	30 (12.8%)	33 (13.4%)	44 (20.3%)	0.103
Hyperlipidemia	42 (18.9%)	32 (13.7%)	29 (11.7%)	30 (14.0%)	0.159
Thyroid	8 (3.6%)	4 (1.7%)	4 (1.6%)	12 (5.6%)	0.046
Inflammatory bowel disease	1 (0.4%)	3 (1.3%)	1 (0.4%)	1 (0.5%)	0.586
Asthma	82 (39.8%)	83 (38.6%)	74 (32.6%)	46 (23.4%)	0.002

Data are presented as number (%) or mean ± SD. GERD, gastroesophageal reflux disease.

**Table 3 jcm-12-06643-t003:** Comparison of COPD drug prescription status according to cluster analysis groups.

	Chronic Bronchitis (N = 224)	Emphysema (N = 235)	Young Smokers (N = 248)	Near Normal (N = 217)	*p*-Value
Drug	191 (88.8%)	200 (90.5%)	212 (91.8%)	173 (84.0%)	0.056
ICSs	4 (1.9%)	2 (0.9%)	1 (0.4%)	0 (0.0%)	0.163
LABAs	27 (12.6%)	43 (19.5%)	29 (12.6%)	25 (12.1%)	0.081
LAMAs	113 (52.6%)	109 (49.3%)	129 (55.8%)	83 (40.3%)	0.009
LABAs + LAMAs	24 (11.2%)	29 (13.1%)	36 (15.6%)	32 (15.5%)	0.481
ICSs + LABAs	73 (34.0%)	75 (33.9%)	64 (27.7%)	42 (20.4%)	0.005
PDE4 inhibitor	13 (6.0%)	8 (3.6%)	6 (2.6%)	6 (2.9%)	0.225
Methylxanthine	57 (26.5%)	76 (34.4%)	58 (25.1%)	47 (22.8%)	0.040

Data are presented as number (%) or mean ± SD. ICSs, inhaled corticosteroids; LABAs, long-acting beta-2 agonists; LAMAs, long-acting muscarinic antagonists; PDE4, phosphodiesterase 4.

**Table 4 jcm-12-06643-t004:** Comparison of exacerbation and mortality according to cluster analysis groups.

	Chronic Bronchitis (N = 224)	Emphysema (N = 235)	Young Smokers (N = 248)	Near Normal (N = 217)	*p*-Value
Moderate exacerbation	68 (30.4%)	86 (36.6%)	72 (29.0%)	52 (24.0%)	0.033
Severe exacerbation	17 (7.6%)	14 (6.0%)	10 (4.0%)	7 (3.2%)	0.153
Moderate exacerbation (frequency)	0.7 ± 1.4	0.8 ± 1.6	0.5 ± 1.1	0.5 ± 1.3	0.019
Severe exacerbation (frequency)	0.1 ± 0.3	0.1 ± 0.6	0.0 ± 0.2	0.0 ± 0.2	0.146
Death	2 (0.9%)	9 (3.8%)	1 (0.4%)	2 (0.9%)	0.009

Data are presented as number (%) or mean ± SD.

**Table 5 jcm-12-06643-t005:** Multiple logistic regression analysis of risk factors for exacerbation of COPD in patients with mild to moderate airflow obstruction.

	Odds Ratio	95% Confidence Intervals	*p*
Young smokers cluster	1.704	0.638–4.688	0.292
Emphysema cluster	2.834	0.893–9.237	0.078
Chronic bronchitis cluster	2.887	1.065–8.192	0.040
Post-bronchodilator FVC, % predicted	1.007	0.978–1.036	0.661
Functional residual capacity, % predicted	1.023	1.007–1.040	0.007
Past history of asthma diagnosis	1.527	0.549–4.045	0.402
6 min walk distance, m	0.999	0.996–1.003	0.693
White blood cells, ×10^3^/mm^3^	1.133	0.953–1.351	0.156
Monocyte, %	0.922	0.776–1.084	0.336
Albumin, g/dL	0.470	0.149–1.440	0.190
Fibrinogen, mg/dL	1.004	1.001–1.008	0.015
Osteoporosis	0.471	0.045–3.120	0.470
Gastroesophageal reflux diseases	2.646	1.142–6.181	0.023

## Data Availability

Datasets cannot be made available to the public. Requests to access these datasets should be directed to the corresponding author.

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
