# Peer review of "Subtypes of Patients with Mild to Moderate Airflow Limitation as Predictors of Chronic Obstructive Pulmonary Disease Exacerbation"

_jcm, 2023, doi:10.3390/jcm12206643_

Round 1
Reviewer 1 Report
Similar studies are present, and possibly by the same lab, on very similar work. The conclusion remarks are of lesser importance as they pertain to basic rationalization for some of the preventive measures mentioned such as cessation of smoking. Lack of study of naive or non-smoking individuals as many near normal individuals also suffered moderate exacerbation. There should be a comparison between native individuals too. This would help to check for native numbers for most of the parameters that are considered. Same could be said for comorbidities. Explanation is needed, why near normal individuals show comorbidities in the range of other groups. A native comparison with healthy individuals in these parameters is a true control and a true baseline parameter measure. compare table one to native individuals (no smoking at all). Also, visual images for various stages of illness and disease would be a great tool for the research fraternity. This could be a bridge between the sample being of a particular region as the visuals can be a help worldwide.
English might need minor corrections.
Author Response
Thank you for your good comments. However, our study was not aimed at comparison with normal controls. The goal of our study was to reveal the heterogeneity that exists in patients with mild and moderate COPD through cluster analysis.
Although the near-normal group in our study had a mean FEV1 above 80%, it was clearly the COPD patients with postbronchodilator FEV1/FVC <0.7. And, given that their average age was 72 years and that more than 70% of them were former or current smokers, their rates of comorbidities or acute exacerbations do not seem at all surprising or unusual.
Additionally, unfortunately, our COPD cohort (KOCOSS) did not separately recruit a control group, so comparative analysis with the normal control group is not possible.

Reviewer 2 Report
Interesting work, I would like to thank the authors for this contribution. The study delivers useful insights in general. The methodology is largely clear, and the manuscript is well-written as well. It is also reassuring to see that the authors have acknowledged potential limitations. However, I would like to offer some suggestions for improvement in the next version, please.
(1)
While the general topic of the study may be clear, there is a need for additional clarification on the specific motivations and goals of the research in the introduction. By providing more context and detail on these aspects, the authors can help readers to understand the relevance and the potential implications of the study's findings.
(2)
The introduction should benefit from referencing previous studies that have utilized K-Means clustering to learn groups or patterns in patient data. For example:
https://doi.org/10.1145/3014812.3014874
(3)
Please provide more details on how the KOCOSS dataset was acquired.
(4)
The sample size of 924 participants is relatively small, which may limit the generalizability of the findings. Future work could aim to include larger and more diverse samples to improve the external validity of the findings. This could be mentioned as part of the limitations.
(5)
Please mention any Machine Learning libraries used, and cite their references.
(6)
The conclusions section should be extended further to include more conclusive remarks about the study.
The quality of language is generally good.
Author Response
Interesting work, I would like to thank the authors for this contribution. The study delivers useful insights in general. The methodology is largely clear, and the manuscript is well-written as well. It is also reassuring to see that the authors have acknowledged potential limitations. However, I would like to offer some suggestions for improvement in the next version, please.
(1) While the general topic of the study may be clear, there is a need for additional clarification on the specific motivations and goals of the research in the introduction. By providing more context and detail on these aspects, the authors can help readers to understand the relevance and the potential implications of the study's findings.
Answer> The specific motivation and goals of the study were written in the introduction. (lines 78-81)
(2) The introduction should benefit from referencing previous studies that have utilized K-Means clustering to learn groups or patterns in patient data. For example:
https://doi.org/10.1145/3014812.3014874
Answer> In the introduction, we referenced previous research using clustering. (lines 75-77)
(3) Please provide more details on how the KOCOSS dataset was acquired.
Answer> The method of acquiring the KOCOSS data set is added in Methods. (lines 90-93)
(4) The sample size of 924 participants is relatively small, which may limit the generalizability of the findings. Future work could aim to include larger and more diverse samples to improve the external validity of the findings. This could be mentioned as part of the limitations.
Answer> The limitations of the study were added in discussion. (lines 269-276)
(5) Please mention any Machine Learning libraries used, and cite their references.
Answer> We mentioned the ‘NbClust’ library was used to determine the appropriate number of clusters for clustering, and cite the references. (lines 130-132)
(6) The conclusions section should be extended further to include more conclusive remarks about the study.
Answer> We expanded conclusion to include definitive opinions. (lines 278-281)

Reviewer 3 Report
1. Add more keywords.
2. Modify [1][2] in Line 45. The link in reference 2 is not working. The style of references 2 and 3 is completely wrong.
3. The GOLD version that you cite is old (2018), there is an update semi annually, have a look at GOLD 2023.
4. In the first paragraph of methods, you cited reference 12 after talking about the inclusion criteria and the diagnosis criteria. You can still mention ....similar to a criteria mentioned in a previous (your own) study, but the diagnosis criteria of COPD is not depending on this study, I mean you need to cite more suitable reference like GOLD.
5. What is the rational of excluding "with measured post-bronchodilator FEV1 values of <50%"?
6. Table 1: you reported p values as 0.000 many times, although this appear in SPSS, when reporting, this is not accepted in statistics, instead write it as <0.001
7. What you mean exactly with "near normal group"?
8. Table 3: write a name for the first column, what you mean with the first raw "Drug" ?
| Drug | 191 (88.8%) | 200 (90.5%) | 212 (91.8%) | 173 (84.0%) | 0.056 |
9. Uniform the decimal points of p values, for example methylxanthine 0.04
10. methylxanthineis not indicated in COPD, however, high percentage were taking it, comment on discussion.
11. The first paragraph in discussion is from the Journal template; delete
12. The discussion should be improved, also, consider comparing with the heterogeneity in asthma and asthma COPD overlap (ACO).
13. The conclusion is not clear, which group is associated with the worst outcome?
14. Finally, I can not find any significant novelty or addition to the current evidence from this study.
Improvement is necessary
Author Response
- Add more keywords.
Answer> By adding keywords, there are more than 5. (lines 40)
- Modify [1][2] in Line 45. The link in reference 2 is not working. The style of references 2 and 3 is completely wrong.
Answer> The style of references 2 and 3 has been corrected, and the links have been checked. (lines 308-310)
- The GOLD version that you cite is old (2018), there is an update semiannually, have a look at GOLD 2023.
The updated 2023 GOLD was cited instead of the older version, 2018. (lines 49)
- In the first paragraph of methods, you cited reference 12 after talking about the inclusion criteria and the diagnosis criteria. You can still mention .... similar to a criteria mentioned in a previous (your own) study, but the diagnosis criteria of COPD is not depending on this study, I mean you need to cite more suitable reference like GOLD.
Answer> Thank you for your comment. For COPD criteria, GOLD was added as a reference. (lines 88)
- What is the rational of excluding "with measured post-bronchodilator FEV1 values of <50%"?
Answer> This is to exclude the severe COPD group with FEV1 value less than 50.
- Table 1: you reported p values as 0.000 many times, although this appear in SPSS, when reporting, this is not accepted in statistics, instead write it as <0.001
Answer> In Table 1, the p value was changed to <0.001, which is acceptable in statistics, instead of 0.000.
- What you mean exactly with "near normal group"?
Answer> The reason they were named the 'near-normal group' is because their mean FEV1 was above 80% of the normal predicted value. (lines 163-164)
- Table 3: write a name for the first column, what you mean with the first raw "Drug" ?
Answer> The meaning of the Drug row refers to COPD patients who have been prescribed any one of the above drugs, such as ICS, LABA, LAMA,, LABA+LAMA, ICS+ LABA, PDE4 inhibitor, and Methylxanthine.
- Uniform the decimal points of p values, for example methylxanthine 0.04
Answer> The decimal point of the p value was unified to the third place below the decimal point
methylxanthineis not indicated in COPD, however, high percentage were taking it, comment on discussion.
Answer> Although methylxantine is not indicated for COPD, it is commonly used in Korea, so so this data also inclued in analysis (lines 274-276)
- The first paragraph in discussion is from the Journal template; delete
We deleted it.
- The discussion should be improved, also, consider comparing with the heterogeneity in asthma and asthma COPD overlap (ACO).
Answer> Your suggestions are appreciated. However, since asthma or asthma-COPD overlap subtype did not appear in our results, it does not seem appropriate to discuss.
- The conclusion is not clear, which group is associated with the worst outcome?
Answer> COPD heterogeneity exists in patients with mild to moderate airflow obstruction, and “chronic bronchitis” phenotype showed high risk of exacerbation. (279-282)
- Finally, I can not find any significant novelty or addition to the current evidence from this study.
Answer> It is difficult to predict acute exacerbations in patients with mild to moderate COPD. In this study, we attempted to analyze phenotypes prone to acute exacerbation through grouping considering a multidimensional variable called k means clustering.

Round 2
Reviewer 1 Report
Data is well presented and explained. Questions are addressed.
Reviewer 3 Report
Most of the comments are addressed
Some improvements are recommended